# The Do's and Don'ts of Grad-CAM in Image Segmentation as demonstrated on the Synapse multi-organ CT Dataset

**Syed Nouman Hasany**                              SYED-NOUMAN.HASANY@UNIV-ROUEN.FR
**Fabrice Mériaudeau**                              FABRICE.MERIAUDEAU@U-BOURGOGNE.FR
**Caroline Petitjean**                              CAROLINE.PETITJEAN@UNIV-ROUEN.FR

## Abstract

The field of eXplainable Artificial Intelligence (xAI) has drawn an immense interest over the last decade. This interest, however, is almost exclusively focused on image classification, whereas other computer vision domains have remained relatively neglected. Recently, however, methods developed in the context of image classification have been extended to other domains such as image segmentation. One such method is Seg-Grad-CAM which is an extension of Grad-CAM. The present paper aims to highlight some of the nuances associated with the utilization of Seg-Grad-CAM in order to generate saliency maps for image segmentation, and instead highlights an alternate application, namely investigating information flow, which can better utilize the capabilities of Seg-Grad-CAM and similar methods. Sample demonstration is provided on an image from the Synapse multi-organ CT dataset.

**Keywords:** image segmentation, explainability, interpretability, XAI, Grad-CAM, Seg-Grad-CAM, information flow

## 1. Introduction

Deep Learning's success in computer vision coupled with it being highly non-linear in nature has motivated researchers to develop techniques in order to investigate the behavior of Deep Learning models. The field of eXplainable Artificial Intelligence (xAI) is dedicated to such techniques. An overwhelming majority of research in xAI is focused on image classification with other computer vision problem domains such as image segmentation having received considerably less attention. Recently, however, segmentation has started receiving some attention with the majority of research focused on extending xAI algorithms from classification to segmentation (Couteaux et al., 2019; Dardouillet et al., 2022). One such attempt is Seg-Grad-CAM (Vinogradova et al., 2020) which is the extension of the popular Grad-CAM (Selvaraju et al., 2019) algorithm - proposed in the context of classification - to segmentation. The present work aims at highlighting some of the nuances associated with a naive extension of Grad-CAM to segmentation. This is followed by identifying a potentially useful way in which Grad-CAM can be utilized in the context of segmentation.

## 2. From Grad-CAM to Seg-Grad-CAM

Grad-CAM is a method utilized in order to generate saliency maps in image classification. In this context, saliency maps are supposed to highlight regions in the input image which were potentially useful for the classification model towards arriving at its final decision

for class **c**. Grad-CAM generates a saliency map by taking a linear combination of the activation maps from a specific model layer. This can be written as follows:

$$L_{Grad-CAM}^c = ReLU \left( \sum_k \alpha_k^c \cdot A^k \right) \tag{1}$$

$A^k$ represents the $k$th activation map and $\alpha_k^c$ represents the $k$th linear coefficient which is the global average pooled value of the gradient matrix of the target class's score ($y^c$) with respect to the activation map ($A^k$):

$$\alpha_k^c = GAP \left( \frac{\partial y^c}{\partial A^k} \right) \tag{2}$$

Given that the output of a segmentation model for a given class is a spatial map and not a single value, (Vinogradova et al., 2020) extended Eq. 2 such that in place of the single value $y^c$, a summation of the target class scores in a region of interest ($\mathcal{M}$) was used:

$$\alpha_k^c = GAP \left( \frac{\partial \sum_{(i,j) \in \mathcal{M}} y_{ij}^c}{\partial A^k} \right) \tag{3}$$

## 3. Nuances

Grad-CAM is usually applied to the final convolutional layer preceding the decision making layer of a classifier. This is understandable given the dual facts that representations deriving from this convolutional layer contain an effectively relevant summary of the input image, and the final decision solely proceeds from these representations. The situation is fairly dissimilar in segmentation. For an encoder-decoder based architecture (U-Net (Ronneberger et al., 2015)), there is no single layer which is responsible for both these tasks. Whereas the bottleneck perhaps makes the strongest case as the layer which carries a relevant summary of the input image, it is not the layer which is solely responsible for the final segmentation decision. The layer responsible for this is the convolutional layer preceding the segmentation head, which, however, does not contain a summary of the input image. While bottleneck layer has been a popular choice (Vinogradova et al., 2020), there is little to no justification as to why it is so. A potential alternate has recently been proposed by (Mullan and Sonka, 2022) which aims at utilizing all layers of the decoder in order to generate a saliency map.

Another issue presents itself when an image contains multiple objects belonging to the same category and a saliency map is to be generated for them individually. Due to the presence of global average pooling in Eq. 3, spatial influence of the gradients is not translated to Eq. 1 as it is reduced to a single number ($\alpha_k^c$). This leads to saliency maps of individual objects being considerably similar despite them being spatially distant. A potential solution was proposed by (Wan et al., 2020; Hasany et al., 2023) which aims at using element-wise multiplication in Eq. 1 by replacing $\alpha_k^c$ with $\frac{\partial \sum_{(i,j) \in \mathcal{M}} y_{ij}^c}{\partial A^k}$.

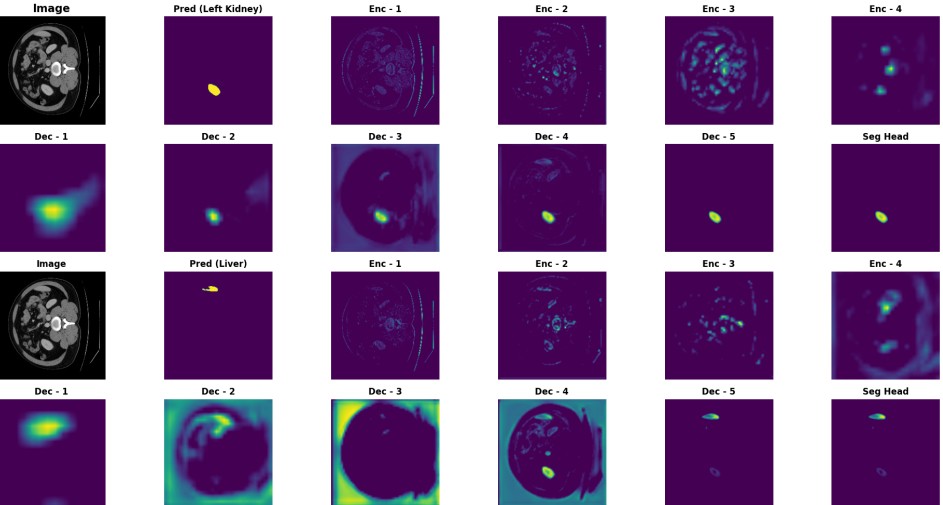

Figure 1: Application of Seg-Grad-CAM to various layers of a U-Net. Two organs (Left Kidney and Liver) are investigated from an image belonging to the Synapse multi-ogran CT dataset. Key: Enc - encoder, Dec - decoder.

## 4. Investigating Information Flow

Instead of utilizing Seg-Grad-CAM to generate saliency maps for segmentation, we believe that it is much more useful to utilize it in order to investigate the segmentation model's representations at various model layers. This can help in deciphering as to how the model makes use of the various representations in order to arrive at the final segmentation decision. Figure. 1 shows the application of Seg-Grad-CAM to explore a U-Net's representations for the predictions of Left Kidney and Liver respectively. In both cases, representations in the encoder stages do not seem to be focused on the organ to segment but appear to be taking the entire image into account in order to construct their respective representations. In the decoder, on the other hand, representations are more focused on identifying the object to be segmented with a gradual refinement in the representations leading to the final segmentation result. An interesting case is that of Liver in the fourth decoder stage where the representations have highlighted not just the Liver and the background, but have also highlighted the Left Kidney. The Left Kidney's representation is then diluted in the fifth decoder stage and the segmentation head in order to arrive at the correct segmentation.

## 5. Conclusion

The present work highlighted some of the nuances associated with a naive extension of Grad-CAM to image segmentation followed by identifying a potential use case for Grad-CAM to image segmentation in the context of exploring information flow.

### Acknowledgments

The authors would like to acknowledge the support of the French Agence Nationale de la Recherche (ANR), under grant Project-ANR-21-CE23-0013 (project MediSEG).

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
