# OpenReview forum: "The Do’s and Don’ts of Grad-CAM in Image Segmentation as demonstrated on the Synapse multi-organ CT Dataset"
_MIDL.io/2024/Short_Papers — MIDL 2024 Short Papers_

### Official Review · Reviewer_bizH · 2024-04-22

**Confidence:** 4
**Final Rating:** 3.5

**Review:**

Summary:
This study explores the use of Seg-Grad-CAM in image segmentation, presenting a nuanced view of its application and suggesting alternative ways to harness its potential, specifically for examining information flow within models.



Pros:
The paper introduces an innovative application of Seg-Grad-CAM for investigating information flow rather than just creating saliency maps, which can provide deeper insights into model behavior.

The paper provides a comprehensive technical breakdown of the limitations when extending Grad-CAM to segmentation, which is crucial for advancing the methodology in xAI.

The paper uses empirical evidence from the Synapse multi-organ CT dataset to illustrate points, enhancing the credibility and relevance of the findings.



Cons:
The paper could strengthen its argument by including quantitative evaluations or comparisons with other methods to validate the effectiveness of the proposed use of Seg-Grad-CAM.

There is limited discussion on whether the findings are applicable to other datasets or segmentation tasks outside of the medical imaging context, which could affect the generalizability of the conclusions.

The choice of layers within U-Net for applying Seg-Grad-CAM lacks a robust justification, potentially undermining the methodological soundness.

Figure 1 could benefit from clearer labels or legends to aid in understanding without referring back to the text.

---

### Decision · Program_Chairs · 2024-04-26

Accept